# Management of Dental Disease in Aardvarks (*Orycteropus afer*) and Potential Use of Cone-Beam Computed Tomography

**DOI:** 10.3390/ani12070845

**Published:** 2022-03-28

**Authors:** Jane E. Christman, Daniel VanderHart, Ben Colmery, Joy Thompson, Ann E. Duncan, Wynona C. Shellabarger

**Affiliations:** 1Detroit Zoological Society, 8450 W 10 Mile Rd., Royal Oak, MI 48067, USA; jthompson@dzs.org (J.T.); aduncan@dzs.org (A.E.D.); wshellabarger@dzs.org (W.C.S.); 2VetRad, Veterinary Teleradiology Interpretation Service, 1321 Centerview Circle, Akron, OH 44321, USA; vanderhart31@yahoo.com; 3Dixboro Veterinary Dental Center, 5740 Plymouth Rd., Ann Arbor, MI 48105, USA; bhc3dvm@aol.com

**Keywords:** aardvark, *Orycteropus afer*, dental pathology, dental imaging, cone-beam computed tomography, CBCT

## Abstract

**Simple Summary:**

Oral disease involving teeth is a common condition of aardvarks (*Orycteropus afer*) housed in zoos, although research is limited. Medical records of aardvarks housed at a single zoo in the United States between 1995 and 2021 were evaluated for dental abnormalities. Eight out of ten aardvarks had dental abnormalities, with most cases having mild disease (4/8). Three patients required the removal of abnormal teeth for treatment. Cone-beam computed tomography (CBCT) is a technology used to create a three-dimensional image of the teeth and skull and was used in two aardvarks. This technology was considered helpful in documenting dental abnormalities in patients with severe disease and may be a valuable tool for veterinarians managing oral disease in aardvarks.

**Abstract:**

Oral disease involving teeth is a common cause of morbidity in aardvarks (*Orycteropus afer*) under managed care. Cases can be challenging due to the species’ unique skull and dental anatomy and limited veterinary literature. A retrospective evaluation was performed on dental examinations in nine aardvarks housed at a single zoological institution in the United States between 1995 and 2021. The prevalence of dental disease in this population was 88%, with most cases categorized as mild (4/8). Clinical signs were only seen in three cases. Facial swelling prior to surgery was the most common clinical sign (3/8). Dental pathology was more common in the mandibular teeth (27/38) compared to the maxillary teeth (11/38). Dental abnormalities found upon intraoral examination included the presence of dental points (7/8), crown elongation (3/8), purulent material within the oral cavity (4/8), loose teeth (2/8), periodontal pockets (2/8), and oronasal fistula (1/8). Three patients required dental extractions with a lateral buccostomy approach. Diagnostic imaging was performed in most cases (7/8), with two cases undergoing cone-beam computed tomography (CBCT) to characterize dental pathology that was difficult to fully evaluate with standard radiography. Tomographic findings are described in both cases. CBCT was found to be a helpful tool for diagnosing and characterizing dental disease in aardvarks.

## 1. Introduction

Aardvarks *(Orycteropus afer*) are large afrotherian mammals and the only extant member of the order Tubulidentata. They are native to sub-Saharan Africa, where they are considered a keystone species [1,2,3]. While uncommon, small numbers of aardvarks are maintained in zoological facilities, with only 33 individuals housed at American Zoological Association (AZA) accredited institutions across 20 institutions in the United States [4]. Literature regarding veterinary care of aardvarks is scarce [5,6,7,8,9,10], and thus management of the aardvark patient can be challenging for zoo practitioners.

Oral disease involving teeth is a common cause of morbidity in aardvarks under managed care [2,6,11,12]. While the prevalence is unknown, it is suspected that a large percentage of aardvarks will have dental disease diagnosed during their lifetimes. The etiology is likely a combination of the unique dental anatomy and dietary considerations. Aardvarks have an elongated skull, a small mouth gape, and a large vermiform tongue, which can make intraoral dental examination challenging. As adults, aardvarks lack incisor and canine teeth, and their cheek teeth are posteriorly positioned in the jaw. The adult dental formula is I(0/0), C(0/0), P(2/2), M(3/3); however supernumerary premolars are commonly identified [1,2,13]. The teeth are uniquely comprised of hexagonal dentin tubules within the pulp cavity, which lack protective enamel and grow continuously throughout life [1,14]. The severity of oral disease involving teeth can range from mild dental malocclusion to osteomyelitis. Moderate or severe disease can be debilitating for the patient and may require repeated anesthesia episodes for treatment [6].

Diagnostic imaging plays an essential role in identifying subgingival pathology and managing dental disease in veterinary practice [15]. Standard and intraoral radiography remain the most common techniques for dental imaging in veterinary medicine, although computed tomography (CT) is increasingly available as technology continues to improve [16,17,18]. Cone-beam computerized tomography (CBCT) is a type of computed tomography that creates an image by emitting a divergent cone beam of radiation and has a long history in dental imaging [19,20,21]. Computed tomography eliminates the superimposition of structures and provides a comprehensive three-dimensional view of the skull and teeth [16,22]. Given the size of the aardvark skull and the challenges of intraoral examination, CT makes an ideal diagnostic tool for dental evaluation in this species. Cost and availability limit access to conventional CT units across many zoo facilities, requiring veterinarians to utilize units at local specialty hospitals, necessitating travel and increasing time under anesthesia. Cone-beam units deliver less radiation exposure to the patient, require fewer building modifications, and provide a cost-effective alternative to conventional CT for applications in zoological facilities [19,20,21,23].

This report reviews cases of dental disease in aardvarks housed at a single zoological institution and describes the feasibility of a portable CBCT unit and tomographic findings in two cases to manage advanced disease.

## 2. Materials and Methods

Medical records of aardvarks under managed care at the Detroit Zoological Society (Detroit Zoo, Royal Oak, MI 48067, USA) from 1995 to 2021 were reviewed retrospectively. Records included dental examinations performed as part of the routine preventative medical protocols or for clinical evaluation of a suspected dental issue. Age, sex, and age at first documentation of dental abnormalities were recorded. Clinical signs prior to dental examination were assessed, as well as the location and type of dental and periodontal lesions noted on intra-oral dental examination and diagnostic imaging. Based on examination findings, cases were categorized as mild, moderate, or severe. Mild cases had crown abnormalities only, while moderate cases had crown abnormalities affecting gingiva or causing malocclusion, and severe cases had evidence of bony lysis upon dental imaging.

Aardvarks undergoing examination were transported from their primary holding areas to the on-site veterinary hospital for examination and diagnostic imaging. Clinical examination was performed in all cases under general anesthesia. Anesthesia was induced using varying drug combinations delivered via intramuscular injection with the most common protocol typically including dexmedetomidine (0.02 mg/kg, Zoetis, Kalamazoo, MI 49007, USA), ketamine (5 mg/kg ZooPharm LLC, Windsor, CO 80550, USA), and midazolam (0.1 mg/kg, WestWard, Eastown, NJ 07724, USA). After 2017, the anesthetic protocols more commonly included medetomidine (0.03 mg.kg, ZooPharm LLC), butorphanol (0.198 mg/kg, Akron Inc., Lake Forest, IL 60045, USA), and midazolam (0.29 mg/kg, ZooPharm LLC). The aardvarks received supplemental oxygen and were maintained on isoflurane delivered via endotracheal intubation or facial mask. Standard radiography of the skull was performed with approximately 45 degrees right and left lateral oblique views using an in-house digital radiography machine. Extractions, when necessary, were performed using a lateral buccostomy approach by a board-certified veterinary dentist with an extensive history working with this species.

Cases undergoing CBCT were positioned in ventral recumbency. The anterior portion of the skull was placed centrally within the gantry, with a scanning range set cranially just posterior to the nasal planum and caudally at the level of the pinnae (Figure 1). The CBCT unit used in all instances was a VetCat portable cone-beam machine (Xoran technologies, Ann Arbor, MI 48108, USA). Scans were performed with the settings 120.0 kVp, 57.6 mAs, with a slice thickness of 0.4 mm. Images were reviewed by a board-certified radiologist.

## 3. Results

### 3.1. Overview of Cases of Dental Disease in Aardvarks Housed at a Single Zoological Institution

Of the ten total aardvarks housed at the zoo during the study period, one was excluded due to incomplete records. Cases included three males and six females, and a total of 46 dental examinations were performed. Eight cases (8/9) had some degree of dental abnormality identified for an overall 88% prevalence within the studied population. These cases are summarized in Table 1. Age at the time of first documentation of dental abnormalities ranged between 2 and 24 years of age, with an average age of 9.5 years (median age 7.5 years). Clinical signs included ptyalism (1/8), facial swelling (3/8), and nasal discharge (1/8). Dental abnormalities found upon intraoral examination included the presence of dental points (7/8), crown elongation (3/8), purulent material within the oral cavity (4/8), loose teeth (2/8), periodontal pockets (2/8), and oronasal fistula (1/8). Radiographic findings included apical lysis (4/8) and crown elongation (3/8).

Diagnoses of dental abnormalities include cheek teeth malocclusion (8/8), caries-like lesions (2/8), periodontal disease (4/8), and alveolar bone loss (1/8). The majority of the cases were categorized as mild, having mild point formation or crown elongation (4/8), one patient was classified as moderate with the presence of moderate point formation causing mucosal ulceration and purulent material upon intraoral examination, and three cases were categorized as severe with severe malocclusion, and alveolar bone loss.

The treatment of cases included occlusal adjustment using a rotating dental burr via intraoral route, performed on 7/8 cases. Surgical closure of the oronasal fistula was performed in case 1. Three cases underwent dental extractions. One case required repeated extractions with a total of eight extracted teeth across four separate dental procedures.

A total of 38 individual teeth had abnormalities cataloged in the dental record. The mandibular arcade was more commonly affected (*n* = 27) compared to the maxillary arcade (*n* = 11). The right mandibular arcade had the highest number of abnormalities (*n* = 14), followed by the left mandibular arcade (*n*= 13), left maxillary arcade (*n* = 11), and right maxillary arcade (*n* = 2). The molars were more commonly affected (*n* = 27) compared to the premolars (*n* = 11).

Diagnostic imaging was performed on the majority of cases (7/8). The most common form of diagnostic imaging was standard radiography of the skull with right and left lateral oblique projections (5/8). Intraoral dental radiography was performed in two cases (2/8). CBCT was performed on two cases, with a total of five scans.

### 3.2. Feasibility of Cone-Beam Computed Tomography (CBCT) and Tomographic Findings

Three CBCT scans were performed in case one, and two scans were performed in case two. Signalment of cases and dates of scans are included in Table 1. Tomographic findings are included in the summaries below. Scans occurred approximately one year apart in each case. All scans showed adequate diagnostic quality and included all relevant anatomy.

#### 3.2.1. Case 1

Cone-beam computed tomography was first performed in October 2019 (Figure 2, Image A). At this time, this patient already had a significant dental history requiring extractions in April of 2012, December of 2014, July of 2015, and August of 2017 (See Table 1, Case 1). CBCT showed three areas of apical lysis identified in the left and right maxillary third molars. A diagnosis was made of alveolar bone loss, secondary to periapical disease and severe malocclusion. No clinical signs were appreciated at that time. Upon intraoral examination, purulent material was associated with the affected teeth. Follow-up CBCT was performed in September of 2020, and moderate progression of the apical lysis was appreciated on the same tooth (left and right maxillary third molar), with the identification of a root remnant present on the right mandibular first molar (extracted in 2012). This root remnant was not observed in the CBCT scan performed in 2019, or in standard skull radiographs performed concurrently at the time of examination. A diastema was noted upon intraoral examination in this region, but no other radiographic or intraoral pathology was identified, and CBCT was considered helpful in the identification of the root remnant. Follow-up CBCT was again performed in November of 2021 (Figure 2, Image B), showing additional root remnants present in the left mandibular first and second premolars and second molar (extracted in 2017) and a missing third molar. These lesions were associated with alveolar bone loss, as well as progressive severe malocclusion. A diagnosis of progressive alveolar bone loss was provided. The remaining tooth on the left mandibular arcade (the first molar) had an elongated crown and was rostrally displaced with a lingual point. Horizontal bone loss was appreciated in the right mandibular third molar. Caries-like lesions (focal loss of attenuation in the crown) were observed on the scan observed in 2021 and were not observed on earlier CBCT scans. CBCT was considered clinically helpful in documenting the progression of tomography lesions in this patient.

#### 3.2.2. Case 2

Cone-beam computed tomography was first performed in August of 2020. At the time of examination, the patient had developed facial swelling along the left lateral maxillary arcade. Concurrent oblique skull radiographs were performed at the time of CBCT. Findings on CBCT and radiographs showed alveolar bone loss around the left maxillary second premolar and first and second molar with crown elongation and point formation. A diagnosis of severe cheek teeth malocclusion and alveolar bone loss was made. Points were filed intraorally using a rotating dental burr. Extractions of the affected teeth were performed in June of 2021. Facial swelling resolved shortly after. CBCT was considered clinically helpful in planning for extractions and characterizing the severity of alveolar bone loss compared to standard radiographs. Follow-up CBCT was completed in September of 2021 during a routine examination. Root remnants were appreciated on the left maxillary second premolar, first, and second molar (leftover from extractions one year prior). No progression in the alveolar bone loss was appreciated. The right maxillary third molar possessed a buccal point. There was progressive crown elongation of the right mandibular first and second molars associated with the extracted teeth. Caries-like lesions (focal loss of attenuation within the crown) were noted in the right mandibular third molar and root remnant of the right maxillary third molar. The patient was clinically doing well, so no further extractions of the teeth with caries lesions were performed, and malocclusion was corrected with a rotating dental burr at the time of examination. The patient is doing well up to the time of manuscript preparation, and CBCT was considered clinically useful for follow-up and lesion identification post-extraction in this patient.

## 4. Discussion

Oral disease involving teeth remains a significant cause of morbidity in zoo-housed aardvarks. The cases identified in this study showed a high prevalence of dental disease in aardvarks as young as two years of age and a relatively high proportion of aardvarks requiring extractions for management. A conference proceeding by Jennifer Langan in 2003 discussed the diagnosis and management of aardvark dental disease in a zoo setting [6]. The high prevalence of dental disease in our study is consistent with Dr. Langan’s experience and is likely representative of the managed population. Further, while mild and moderate dental disease frequently occurs in managed aardvarks, our study suggests a smaller proportion of cases having severe disease. This may be secondary to the age of the aardvarks housed at the particular institutions, genetic variation, or differences in management practices. Improvements in the collective knowledge and management of aardvarks over the past two decades may have also contributed to changes in disease prevalence and severity. Longitudinal studies with an increased sample size would be helpful to determine the prevalence and severity of this disease in all aardvarks across zoological institutions in the United States and abroad.

The majority of dental abnormalities were identified on routine physical examination with no clinical signs or suspicion of progressive dental disease. Clinical signs in affected animals were generally considered mild and included ptyalism, facial swelling, and nasal discharge. Facial swelling was the most encountered clinical sign but only noted in a few instances where dental pathology was already documented in prior examinations. In general, aardvarks with facial swelling had evidence of more advanced disease. These cases highlight the importance of routine preventative dental examinations and dental imaging for the veterinary care of aardvarks housed at zoological institutions. This is especially true with a managed population that is becoming more aged, as oral disease involving teeth is more common in older animals. Early diagnosis and treatment aid in improving patient welfare and can result in significantly improved outcomes [24].

The two cases where CBCT was used as a management tool had complicated dental histories that required extractions. This technology was thought to be instrumental in identifying apical pathology that was unnoticed or challenging to see on intraoral examination or standard radiographs. The comprehensive nature of the images and the ability to perform three-dimensional reconstruction were considered helpful to map out lesions, especially in a species with challenging and variable dental anatomy. The ability to repeat CBCT scans during subsequent examinations allowed for better patient follow-up and clinical management for both cases.

Both cases who underwent CBCT noted evidence of root-remnant formation post-extraction. The molars appeared to develop four years after the extraction was performed in one case. These molars were not noted in earlier imaging, indicating root formation long post-extraction. All teeth were considered completely extracted at the time of the procedure. This highlights the challenges of tooth extraction in aardvarks for veterinary practitioners and may also suggest a propensity of this species for root retention post-extraction. The reason for this is unknown and may be secondary to the preservation of the radix in the growth center post-extraction and the unique composition of the tooth. Root remnants can promote inflammation and become a nidus for infection [25,26]. Aardvark patients who undergo dental extractions should be monitored for root-remnant formation, and the use of CBCT may be helpful for the identification of these lesions.

A limitation of this study is the small sample size and retrospective nature of data collection. The small size is due to the small numbers of aardvarks housed at a single institution and is representative of the small population of zoo-managed aardvarks in AZA institutions across the United States. Because of this, direct comparisons of the diagnostic yield of conventional radiographs and CBCT cannot be performed as concurrent images were not routinely obtained to minimize time under anesthesia for the patient. While objective comparisons between the two techniques cannot be reported, the purpose of this study is to describe the feasibility and use of this technique and incorporation of CBCT in a zoological practice for the management of oral disease involving teeth in the aardvark patient.

The CBCT unit used for the described cases is a veterinary-specific machine designed for dental imaging. It has a limited gantry size, which can thus impede the scanning range for species with larger skull sizes or other anatomy that precludes positioning. The cases included in this study were easily obtained, and scans included all relevant anatomy; however, this required care when positioning. Newer models of CBCT in the veterinary dental market will hopefully improve these limitations, increasing the feasibility of this technology for a broader range of uses in zoological settings.

While CBCT creates high-quality images of skeletal anatomy, it provides poor detail of soft tissue structures. This provides limited detail of surrounding structures within the skull, including ocular and pharyngeal anatomy. Other differentials for facial swelling across species include trauma, cellulitis, foreign body presence, and neoplasia. Therefore, in a patient with facial swelling, further diagnostics may be necessary to characterize other causes of soft tissue disease that may be present, such as ultrasound or magnetic resonance imaging (MRI).

Another option for imaging dental lesions is using dental radiography units. These provide a two-dimensional image of single or multiple teeth and have long been used in human and veterinary dentistry [15]. They are widely available and already routinely incorporated in many zoological practices. However, for aardvark patients, positioning the small plate intraorally remains challenging due to the limited ability to open the mouth and the caudal positioning of the cheek teeth. This methodology was attempted in two cases and could only reach the more rostral premolars. Furthermore, it is the authors’ opinion that the large tooth size in adult aardvarks generally precludes the use of the small dental plate sizes, limiting the comprehensive view of a single tooth.

The etiology of oral disease involving teeth in aardvarks remains unknown. In equine dentistry, acquired dental disease can occur secondary to age-related changes and attrition, which may also be present in the zoo-managed aardvark population [27]. Speculation exists that diets offered under managed care may play a role [6]. In their natural environment, aardvark diets are high in insect matter, consisting primarily of termites, and as such, are high in protein [28]. The diet of aardvarks in zoo populations is an extruded small pellet diet with supplemented produce and insect material designed to mimic the nutritional components of the wild-type diet and is high in protein and low in carbohydrates. The dissolvable pelleted diet, however, may become trapped between teeth and contribute to the development of periodontitis [6]. While zoos have come a long way in adjusting the aardvark diet nutritionally, further adjustments may still be necessary to further reduce the prevalence of dental disease in managed populations. Traumatic etiologies, such as excessive tooth grinding or inappropriate wear to the occlusal surface of teeth, can also be considered. Likely, the pathophysiology of aardvark dental disease is multifactorial and a potential area of future study.

## 5. Conclusions

Oral disease involving teeth continues to be a disease of concern for aardvarks under managed care. The use of cone-beam computed tomography (CBCT) is a feasible tool that may help identify dental lesions in zoo-managed aardvarks, thereby improving clinical management and patient welfare.

## Figures and Tables

**Figure 1 animals-12-00845-f001:**
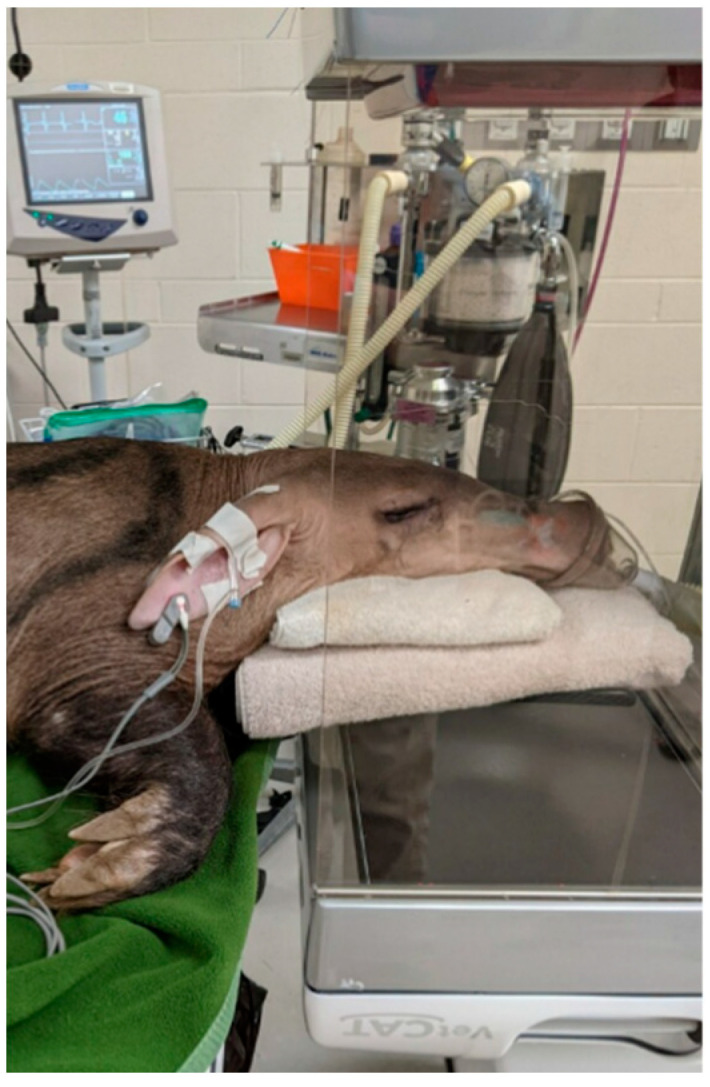
Photograph of aardvark under general anesthesia maintained on a face mask with the anterior portion of the head placed in the gantry of the cone-beam computed tomography (CBCT) unit. This image highlights the scanning range of the unit.

**Figure 2 animals-12-00845-f002:**
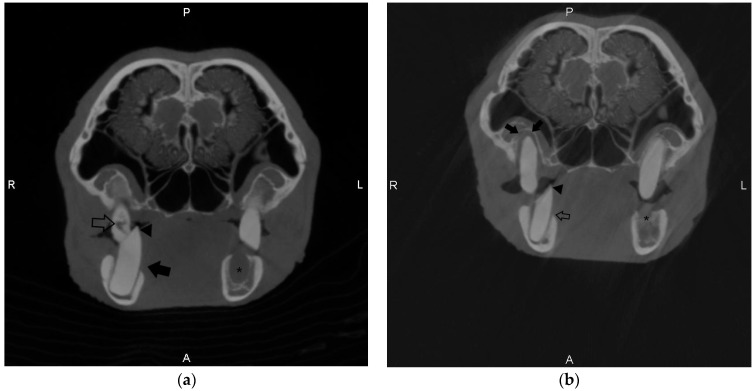
Transverse images of the skull of Case 1 using cone-beam computed tomography (CBCT) taken at two time points. Images were captured at the level of the third molar tooth. (**a**) Taken in 2019 and shows horizontal bone loss affecting the right third mandibular molar (black arrow), a large point of the right third mandibular molar (arrowhead), caries-like lesions associated with the crown of the right third maxillary molar tooth (open arrow), and a missing left third mandibular molar tooth with an associated alveolar defect (Asterix). (**b**) Taken in 2021 and shows significant alveolar bone loss surrounding the tooth root of the right third maxillary molar (solid arrows) and continued horizontal bone loss associated with the right third mandibular molar tooth (open arrow). The right third mandibular molar is axially displaced relative to the maxillary molar, and there is a significant point associated with the right third mandibular molar. The left third mandibular molar tooth is absent, and there is an alveolar defect at the level of the missing tooth (Asterix).

**Table 1 animals-12-00845-t001:** Cases of dental disease in aardvarks under managed care at a single zoological institution in the United States, including sex, age of onset of dental disease, clinical and radiographic signs, categorization of the severity of dental disease, and locations of dental extractions (if noted in medical records). Abbreviations used include L (left), R (right), max (maxillary arcade), mandib (mandibular arcade), P (premolar), and M (molar). The subsequent number after P or M designates the number of the corresponding tooth.

Case	Sex	Age of Onset (yr)	Clinical Signs and Diagnoses	Severity of Disease	Extractions
1	Male	8	Ptyalism, Facial swelling, dental points, nasal discharge, fistula formation, purulent material in the oral cavity, periodontal pockets, apical lysis, retained roots.	Severe	Aug 2017: L mandib M 1,2,3; July 2015: L max M2; Dec 2014: R mandib M3; April 2012: L max M3, R mandib PM2 M1;
2	Female	5	Facial swelling, dental points, purulent material in the oral cavity, apical lysis	Moderate	June 2021: L max PM2, M1, M2
3	Female	4	Mild dental points, crown elongation	Mild	None
4	Female	7	Crown elongation, apical lysis, purulent material in the oral cavity, periodontal pockets	Moderate	None
5	Male	2	Mild dental points	Mild	None
6	Male	15	Mild dental points	Mild	None
7	Female	11	Mild dental points	Mild	None
8	Female	24	Facial swelling, dental points, crown elongation, purulent material in the oral cavity, apical lysis	Severe	Jan 2004: L mandib M3

## Data Availability

The data presented in this study are available upon request from the corresponding author. The data are not publicly available due to privacy considerations for the managing institution.

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
