# Peer review of "Management of Dental Disease in Aardvarks (*Orycteropus afer*) and Potential Use of Cone-Beam Computed Tomography"

_animals, 2022, doi:10.3390/ani12070845_

Round 1
Reviewer 1 Report
Aardvarks dental diseases are common condition in captive animals, however, there is a lack of available data in the relevant literature. Tooth extractions, though, have been described on the internet.
Article describes clinical signs caused by dental diseases in aardvarks and applied treatment procedures. Since there is no other causes of observed conditions, and the main advancement is the use of CBCT, I suggest modification of the title.
Aardvarks (Orycteropus afer) dental diseases and management: potential use of Cone-Beam Computed Tomography
Cone-Beam Computed Tomography is a well-known procedure with a long-history of application in dental medicine.
Retrospective part of the study does not contribute to the advances in management of oral diseases and therefore has no special place in this ms. I suggest to include it as an overview of dental problems in aardvarks from this collection in the past. Please, provide diagnosis, and if available, other dental procedures that were applied except extractions.
For Case 1 and Case 2, provide diagnosis and more details about results obtained by the use of CBCT and advances brought up by introduction of this method.
For all, discuss potential causes that have led to these conditions.
Minor:
Results, Line 118 - three males and five females
Discussion, line 207 - reference must be inserted (number)
line 241 - this belongs to the Material and Method section
Lines 238-250 - aardvarks have so called radicular hypsodonty. This will result with regrowth of the radix in growth center is preserved.
Author Response
Dear Reviewer,
Thank you for your time and thoughtful regarding the manuscript entitled “Advances in the Management of Oral Disease Involving Teeth in Aardvarks (Orycteropus afer): A Retrospective of Cases at a Single Zoological Institution in the United States and Use of Cone-Beam Computed Tomography.”. We hope you will agree that we have addressed your comments and concerns appropriately.
We have amended the title of the of the manuscript to “Management of Dental Disease in Aardvarks and Potential Use of Cone-Beam Computed Tomography”, which we hope the reviewers feel better captures the scope of the paper. We have also adjusted the retrospective section as an overview of cases and expanded the diagnoses within that section and expanded the interpretation of the cases in which cone-beam computed tomography was used.
We have listed all comments and suggestions below, with specific actions listed underneath each comment with the new line number for reference.
Please do not hesitate to contact us if additional review or clarification is necessary.
Sincerely,
Drs. Jane Christman, Daniel VanderHart, Ben Colmery, Ann Duncan, Wynona Shellabargar and Ms. Joy Thompson.
Reviewer 1:
Aardvarks dental diseases are common condition in captive animals, however, there is a lack of available data in the relevant literature. Tooth extractions, though, have been described on the internet.
The authors agree that relevant peer-reviewed literature regarding dental disease in aardvarks is extremely limited. It is the hope that this manuscript acts like a pilot study and will spark further publications and advancement regarding this condition in such a rare species to manage in captivity.
Article describes clinical signs caused by dental diseases in aardvarks and applied treatment procedures. Since there is no other causes of observed conditions, and the main advancement is the use of CBCT, I suggest modification of the title.
Aardvarks (Orycteropus afer) dental diseases and management: potential use of Cone-Beam Computed Tomography
The authors thank the reviewer for this suggestion and agree that the title of the article can be misleading regarding the term “advances” and appreciate the suggestion for an alternate title. We have amended the title to a similarly worded “Management of Dental Disease in Aardvarks (Orycteropus afer) and Potential Use of Cone-Beam Computed Tomography”, which we hope the reviewer agrees better captures the scope of the manuscript.
Cone-Beam Computed Tomography is a well-known procedure with a long-history of application in dental medicine.
Line 74: The authors agree that Cone-Beam Computed Tomography has extensive use in dental practice. This paragraph in the introduction has been amended to better reflect the current use of this technology, however, as CBCT use in zoological facilities remains extremely limited, the authors’ felt a description of the technology is still warranted and description of cases is useful.
Retrospective part of the study does not contribute to the advances in management of oral diseases and therefore has no special place in this ms. I suggest to include it as an overview of dental problems in aardvarks from this collection in the past. Please, provide diagnosis, and if available, other dental procedures that were applied except extractions.
Line 130: The authors appreciate this comment regarding the retrospective section. The inclusion was meant to provide additional detail and background, which is limited in the literature regarding this condition, as well as context for the lesions noted on cone-beam computed tomography. The authors clarified the inclusion of this section by changing the title of the section within the results to “overview of cases of dental disease in aardvarks housed at a single zoological institution” and adjusting the terminology from “retrospective examination” to “overview of cases” throughout the manuscript. The section was also separated to include additional information regarding diagnoses and treatments.
For Case 1 and Case 2, provide diagnosis and more details about results obtained by the use of CBCT and advances brought up by introduction of this method.
Line: The authors appreciate this feedback and have expanded both case one and case two to include relevant diagnoses obtained from the CBCT scans and interpretation of usefulness of the scans clinically, which the authors hope will help readers see additional benefit to this technology.
For all, discuss potential causes that have led to these conditions.
Line181-207;219-240: The authors have expanded the last paragraph regarding the etiology of dental disease to address additional potential causes.
Minor:
Results, Line 118 - three males and five females
Line 133: This line has been changed to correct the typographical error of “size”, which was meant to refer to the number “six” rather than “five”, as it was in reference to the total number of aardvarks included (nine total). This has been changed in the manuscript.
Discussion, line 207 - reference must be inserted (number)
Line 213: The reference has been included.
line 241 - this belongs to the Material and Method section
Line 248: The author believes that this correction is in reference to the sentence “All extractions were performed by a board-certified veterinary dentist”. This is clarified in the Materials and Methods section.
Lines 238-250 - aardvarks have so called radicular hypsodonty. This will result with regrowth of the radix in growth center is preserved.
Line 324: This sentence has been clarified to include preservation of the radix in the growth center as a potential etiology of root remnant formation.
Reviewer 2 Report
In my opinion the the title is misleading because it doesn't describe any novel info or "Advance" in the management of oral disease in aardvarks. I was aspected , considering the title, a more focused paper on new management , theraphy ,causes of dental diseases such as much more info of the use of CBCT, normal/abnormal, more detailed pictures and descrisption .
Said that, I find really interesting this paper because it will increase knowledge in zoo medicine but I would suggest to modified the title considering the main part of it is a retrospective study.
In the end, I would not "emphasize" the use of CBCT considering that just two cases were described, but probably could be of valid help in managing dental disease in aardvarks.
Author Response
Dear Reviewer,
Thank you for your time and thoughtful regarding the manuscript entitled “Advances in the Management of Oral Disease Involving Teeth in Aardvarks (Orycteropus afer): A Retrospective of Cases at a Single Zoological Institution in the United States and Use of Cone-Beam Computed Tomography.”. We hope you will agree that we have addressed your comments and concerns appropriately.
We have amended the title of the of the manuscript to “Management of Dental Disease in Aardvarks and Potential Use of Cone-Beam Computed Tomography”, which we hope the reviewers feel better captures the scope of the paper.
We have listed all comments and suggestions below, with specific actions listed underneath each comment with the new line number for reference.
Please do not hesitate to contact us if additional review or clarification is necessary.
Sincerely,
Drs. Jane Christman, Daniel VanderHart, Ben Colmery, Ann Duncan, Wynona Shellabargar and Ms. Joy Thompson.
Reviewer 2:
In my opinion the the title is misleading because it doesn't describe any novel info or "Advance" in the management of oral disease in aardvarks. I was aspected , considering the title, a more focused paper on new management , theraphy ,causes of dental diseases such as much more info of the use of CBCT, normal/abnormal, more detailed pictures and descrisption .
The authors appreciate the reviewers’ comments regarding the title of the paper and have amended it hopefully in a way the reviewer finds agreeable (see below).
Said that, I find really interesting this paper because it will increase knowledge in zoo medicine but I would suggest to modified the title considering the main part of it is a retrospective study.
The authors thank the reviewer for their appreciation of the intent of this paper to increase the knowledge base of zoo medicine in the management of this disease in aardvarks. We have amended the title to “Management of Dental Disease in Aardvarks (Orycteropus afer) and Potential Use of Cone-Beam Computed Tomography”, which we hope the reviewer agrees better captures the scope of the manuscript.
In the end, I would not "emphasize" the use of CBCT considering that just two cases were described, but probably could be of valid help in managing dental disease in aardvarks.
The authors appreciate this suggestion and have amended the wording of the emphasis of the use of CBCT throughout the manuscript. The authors feel that the inclusion of the tomographic findings and description of the two cases can be helpful, however, to veterinarians in zoo practice managing dental disease in aardvark patients and feel that highlighting this technology beyond what is included in the retrospective is useful to the reader.
Reviewer 3 Report
This is a review of “Advances in the Management of Oral Disease Involving Teeth 2 in Aardvarks (Orycteropus afer): A Retrospective of Cases at a 3 Single Zoological Institution in the United States and Use of 4 Cone-Beam Computed Tomography.”
This manuscript is extremely well written and illustrated as well as interesting and pertinent. The only revisions needed are very minor ones of providing further details on lesion description / severity in the abstract and Materials and Methods.
Abstract:
Please list specific abnormalities (fistula, osteomyelitis) identified. You can just copy Lines 125-127 for this purpose.
Line 27 in abstract: please clarify the facial swelling was before surgery. The detail that there were few clinical signs in general are also important to note.
Material and methods”
Clarify which lesions were assessed
That’s it!
Author Response
Dear Reviewer,
Thank you for your time and thoughtful regarding the manuscript entitled “Advances in the Management of Oral Disease Involving Teeth in Aardvarks (Orycteropus afer): A Retrospective of Cases at a Single Zoological Institution in the United States and Use of Cone-Beam Computed Tomography.”. We hope you will agree that we have addressed your comments and concerns appropriately.
We have provided further details, as requested, in the materials/methods and abstract section, which we hope will provide clarification.
We have listed all comments and suggestions below, with specific actions listed underneath each comment with the new line number for reference.
Please do not hesitate to contact us if additional review or clarification is necessary.
Sincerely,
Drs. Jane Christman, Daniel VanderHart, Ben Colmery, Ann Duncan, Wynona Shellabargar and Ms. Joy Thompson.
This is a review of “Advances in the Management of Oral Disease Involving Teeth 2 in Aardvarks (Orycteropus afer): A Retrospective of Cases at a 3 Single Zoological Institution in the United States and Use of 4 Cone-Beam Computed Tomography.”
This manuscript is extremely well written and illustrated as well as interesting and pertinent. The only revisions needed are very minor ones of providing further details on lesion description / severity in the abstract and Materials and Methods.
Abstract:
Please list specific abnormalities (fistula, osteomyelitis) identified. You can just copy Lines 125-127 for this purpose.
Line 27: The dental abnormalities listed in the results section are included in the abstract.
Line 27 in abstract: please clarify the facial swelling was before surgery. The detail that there were few clinical signs in general are also important to note.
Line 27: Ammended this sentence to include that clinical signs were only seen in three cases, and that facial swelling occurred prior to surgery.
Material and methods”
Clarify which lesions were assessed
Line 98: A sentence was included to clarify the type of lesions that were assessed.
Round 2
Reviewer 1 Report
None